# Gelatine Blends Modified with Polysaccharides: A Potential Alternative to Non-Degradable Plastics

**DOI:** 10.3390/ijms25084333

**Published:** 2024-04-14

**Authors:** Oleksandra Dzeikala, Miroslawa Prochon, Natalia Sedzikowska

**Affiliations:** Institute of Polymer and Dye Technology, Faculty of Chemistry, Lodz University of Technology, Stefanowskiego 16, 90-537 Lodz, Poland; oleksandra.dzeikala@dokt.p.lodz.pl (O.D.); 231435@edu.p.lodz.pl (N.S.)

**Keywords:** bioplastic, biopolymers, environment, biodegradation, modification, polysaccharides, gelatine, agar, carrageenans

## Abstract

Non-degradable plastics of petrochemical origin are a contemporary problem of society. Due to the large amount of plastic waste, there are problems with their disposal or storage, where the most common types of plastic waste are disposable tableware, bags, packaging, bottles, and containers, and not all of them can be recycled. Due to growing ecological awareness, interest in the topics of biodegradable materials suitable for disposable items has begun to reduce the consumption of non-degradable plastics. An example of such materials are biodegradable biopolymers and their derivatives, which can be used to create the so-called bioplastics and biopolymer blends. In this article, gelatine blends modified with polysaccharides (e.g., agarose or carrageenan) were created and tested in order to obtain a stable biopolymer coating. Various techniques were used to characterize the resulting bioplastics, including Fourier-transform infrared spectroscopy (FTIR), thermogravimetric analysis (TGA)/differential scanning calorimetry (DSC), contact angle measurements, and surface energy characterization. The influence of thermal and microbiological degradation on the properties of the blends was also investigated. From the analysis, it can be observed that the addition of agarose increased the hardness of the mixture by 27% compared to the control sample without the addition of polysaccharides. In addition, there was an increase in the surface energy (24%), softening point (15%), and glass transition temperature (14%) compared to the control sample. The addition of starch to the gelatine matrix increased the softening point by 15% and the glass transition temperature by 6%. After aging, both compounds showed an increase in hardness of 26% and a decrease in tensile strength of 60%. This offers an opportunity as application materials in the form of biopolymer coatings, dietary supplements, skin care products, short-term and single-contact decorative elements, food, medical, floriculture, and decorative industries.

## 1. Introduction

Recently, about 40% of plastics used are used to produce disposable packaging, which generates waste that pollutes the natural environment. Replacing plastics with biodegradable and biopolymer materials of various origins makes it possible to move away from polymers obtained from petrochemical raw materials. Animal and plant proteins (gelatine, casein, collagen, gluten, soy protein, and zein), polysaccharides (starch, cellulose, chitosan, agar, and carrageenan) or lipids (fats, oils, and waxes) are most often used to produce biopolymer films [1,2]. 

Fibrogenic properties are characterized by a biopolymer of animal origin, such as gelatine. The structure of this protein consists of amino acids, predominantly glycine, proline, and hydroxyproline, which are presented as a three-member sequence connected by peptide bonds. The natural polymer is created after the hydrolysis of the triple helix, which involves the depolymerization of the structure of collagen extracted from animal raw materials [3,4]. The polymer network formed after the denaturation of gelatine has the form of a hydrogel with low stiffness, which swells under the influence of water. The origin of the gelatine and the cross-linking substance used influence the mechanical and thermal properties of the obtained gelatine matrix [5]. In the case of a collagen derivative, low-volatility plasticizing additives are used to improve the functional properties such as elongation at break, elasticity, plasticity, and stiffness. For this purpose, polyols (glycerol, xylitol, etc.), monosaccharides (e.g., glucose and mannose), and fatty acids are used [6,7]. Due to the potential replacement of previously used plastics as disposable food packaging, these materials should be decomposed by composting with the participation of micro-organisms. For example, the creation of edible single-layer films based on gelatine most often involves evaporation from solvent solutions or by dispersing protein [8]. The composition of the mixture to create the gel matrix is as follows: 20–30% gelatine, 10–30% plasticizer (glycerol or sorbitol), and 40–70% water [9]. 

Edible films with a thickness of less than 0.3 mm must consist of food-safe and edible ingredients. In addition, they must fulfill several functions, such as protection against the loss of moisture and important ingredients, or selectively participate in the exchange of gases, e.g., oxygen and carbon dioxide [10,11]. Active food packaging, in turn, aims to extend the shelf life and increase the nutritional value of food by preventing dehydration, oxidative rancidity, and surface browning [12].

Polysaccharides with hydrophilic properties and a crystalline structure are very sensitive to the presence of moisture, but they provide the composites with appropriate mechanical properties. For example, agar, carrageenan, alginate, starch, pectin, xanthan gum, and others can be used to produce biodegradable films [12,13].

Agar is a polysaccharide obtained after processing some species of marine algae of the genus Gracilaria and Gelidium from the group of red algae (Rhydophyta), which have cell walls consisting of layers. Each layer is composed of cellulose microfibrils grouped into macrofibrils. The basic structure of this polysaccharide is two monomers: 3,6-anhydro-L-galactose and D-galactose connected by alternating α-(1,3) and β-(1,4)-glycosidic bonds [14]. Agar forms reversible gels, so it can be used as a thickening and gelling agent for aqueous solutions. During cross-linking, the molecules form double helices that are compatible with gelatine [15]. Agar gel without additives is characterized by a limited resistance to aging (photodegradation), fluctuations in the sol–gel transformation temperature, and a low degree of resistance to moisture. This causes the formation of microcracks and increases the brittleness of the material [16,17]. Nevertheless, it is a compound that can create composites and blends, so it is used as an addition to another biopolymer that creates a gel matrix, e.g., cellulose (as a polysaccharide that allows for a greater resistance to heating and microwave effects, increasing tensile strength and stiffness) [15,18] or gelatine (with stabilizing and emulsifying and thickening protein, with the ability to bind water, and limited thermal stability and mechanical strength) [19].

Carrageenan is a polysaccharide obtained from the red algae species Chondrus crispus, Eucheuma denticulatum, and Gigartina stellate, which, analogously to agar, is isolated from the cell wall and intercellular matrix of marine algae [20]. The chemical structure of carrageenan is a sulfonated (depending on the type of carrageenan, it contains 22–35% sulfonic groups) chain of D-galactose and 3,6-anhydro-L-galactose connected by α-(1,3) and β-(1,4) bonds) glycosides. Depending on the raw material, its lifetime, and its extraction method, a specific type of carrageenan is obtained, differing in the place of substitution and the number of sulfone groups [21]. Carrageenan is a biopolymer that has many important properties from the point of view of its usefulness, i.e., anti-aging, antioxidant, and anti-cancer properties. Moreover, it has the ability to thicken solutions and create reversible gels, with the sol–gel transition temperature depending on the type of carrageenan. Commercial carrageenan comes in the form of sodium, potassium, magnesium, or calcium salts, which have an increased structural stability compared to carrageenan containing free sulfone groups. The number of acid groups, their stabilization by cations, and the presence of other substances affect the viscosity and strength of the formed gel. In addition, the viscosity itself increases with a higher polysaccharide concentration and with decreasing temperature. The disadvantage of carrageenan composites is their sensitivity to a low solution pH and high temperature, as these factors significantly affect the structure through depolymerization [21,22].

Their use as edible, biodegradable disposable food packaging requires appropriately high mechanical, thermal, and moisture resistance properties. K. Wang et al. [23], who investigated the effect of corn starch on the properties of collagen films, noted the roughness of the obtained sample, which did not occur for films without polysaccharide. Starch increased tensile strength when heated and decreased water solubility. Thermal stability, tested using the DSC method, was increased under the influence of polysaccharide. 

R. Kumar et al. [24] used corn starch modified with 5% citric acid and gelatine to obtain biodegradable composite films plasticized with sorbitol, examining the optimal ratio of modified corn starch and gelatine. The composite films were characterized by an increase in thickness with an increase in gelatine content, as well as a hydrophilic character, increased swelling and solubility in water, reduced transparency, and improved mechanical properties. The obtained composite films extended the shelf life of cucumber to 16 days.

The analysis of the relationship between the structure and properties of the gelatine–chitosan composite was performed using scanning electron microscopy by H.R. Naseri et al. [25]. The authors compared the obtained SEM images (Scanning Electron Microscopy) and infrared spectroscopy results, based on which they confirmed the presence of interactions between two polymers (gelatine and chitosan) and the additive (essential oil from the Ferulago angulate plant). The resulting composite had a greater water vapor permeability after adding the oil.

Sodium alginate and agar can form biodegradable, two-layer films enriched with essential oils. The natural oil gives the composite antimicrobial properties, which can potentially be used in biopolymer food packaging. B. Zhang et al. [26] achieved the best properties for the film with the addition of ginger essential oil. The structure of the film was homogeneous and tests confirmed its antimicrobial activity.

A gelatine/agar-based film integrated with a Pickerin emulsion from a solution of cellulose nanofibers and clove oil was obtained by casting by S. Roy and J-W. Rhim [27]. The resulting films had a high UV ultraviolet light barrier, and antioxidant and enhanced waterproof properties. A small addition of Pickerin emulsion slightly influenced the mechanical and barrier properties and gradually increased the film thickness. All gelatine/agar-based films showed multi-stage thermal decomposition. Already pure gelatine–agar foil showed some DPPH (2,2-diphenyl-1-picrylhydrazyl) and ABTS (2,2′-azino-bis(3-ethylbenzothiazoline-6-sulfonic acid) radical scavenging activity (presence of antioxidant peptides in gelatine), and the addition of Pickerin emulsion significantly increased this property.

A. Syarifuddin et al. [28] modified gelatine–carrageenan films with canola oil from Canadian rapeseed (Brassica napus) to produce a hydrophobic dispersion layer. The hydrophobic additive prevented water evaporation, which resulted in the formation of thicker films that dissolved in water to a lesser extent than the composite without this additive. The water vapor transmission rate of hydrophobic composites decreased significantly due to the increase in resistance to the transfer of water particles caused by the change in the polarization of the matrix components. The introduction of oil resulted in a significant reduction in tensile strength, which could have caused structure discontinuities. A higher weight share of κ-carrageenan enabled the formation of interactions between the polysaccharide chains, and the hydroxyl groups used did not form hydrogen bonds with water, which resulted in a lower moisture content of the composites. It was found that the mechanical properties are significantly influenced by the film thickness and the presence of lipids.

The aim of the research work is to enhance gelatine blends by incorporating polysaccharides such as agar and carrageenan, with the goal of positively influencing the properties of the resulting films. The study has examined the film properties for varying amounts of agar and carrageenan additives. The resulting bioplastics are envisioned to be environmentally friendly alternatives that can be utilized as single-use products. These innovative materials hold promise not only in the food packaging sector but also in various other applications.

## 2. Results and Discussion

### 2.1. Preparation of the Blends

Biopolymer blends were obtained by synthesis in a reaction set consisting of a three-necked flask, a thermometer, and a magnetic stirrer in an oil bath at 75 °C (Figure 1). The reaction was completed after all components had dissolved. After the synthesis was completed, the whole was transferred to a vacuum evaporator (name b-401, producer BUCHI) to remove the remaining solvent. An efficiency of 85% was achieved in the process. The use of a three-necked flask may be important for the production of gelatine blends and the resulting yields for several reasons:

Mixing and homogenization: The three-necked flask enables the effective mixing of various ingredients, including gelatine and any additives or fillers. This makes it possible to obtain a homogeneous mixture, which is crucial for obtaining consistent mechanical and structural properties of the films.Process control: As a result of the different flask necks, you can control the production process, e.g., by adding subsequent ingredients at the right moments, regulating the temperature, or mixing time. This allows the process to be adapted to the specific requirements for gelatine blends, which may have a significant impact on the final quality of the product. 

The unformed material was subjected to a processing process of pressing at a temperature of 140–150 °C and under a pressure of 10 MPa in order to give the desired shape—in the case of the research, these were plates with dimensions 100 mm × 100 mm × 1 mm. Subsequently, an element dimensioned for individual examinations and tests was cut from the boards using an automatic die described in Table 1. Table 2 shows the composition of the polymer blends produced.

### 2.2. FTIR Spectroscopy

A Fourier-transform infrared spectroscopy (FTIR) test was performed for all composites to determine the functional groups. The acquired spectra are shown in Figure 2, Figure 3 and Figure 4. 

Figure 2 shows a summary of the spectra of the samples (control, agar (35 parts by weight), and carrageenan (35 parts by weight)). This arrangement allowed for the determination of structural changes in the composites related to the addition of polysaccharides.

Based on the spectra presented in Figure 2, it can be concluded that the blends produced have similar functional groups in their structure. The broad band for the wave number of 3270 cm^−1^ is assigned to Amides A and B. They are attributed to stretching vibrations occurring when a secondary amine group is present in the structure, coupled to vibrations of the primary alkyl amide I, according to the Fermi resonance phenomenon, due to the very similar excitation energies of these vibrations. In the same range of 3200–3600 cm^−1^, the hydroxyl group absorbs, so an overlap of bands is observed. These groups (N-H and O-H) are able to form hydrogen bonds with other molecules containing an oxygen or nitrogen atom, e.g., glycerine or water, which increases the bandwidth from the described range. An approximation of the spectrum in the range of 2800–3000 cm^−1^ indicates the presence of peaks at the wave number of 2916 cm^−1^ and 2850 for the control sample, which are attributed to symmetric and asymmetric CH stretches in aromatic methoxy groups and in methyl and methylene groups of side chains formed from lignin and pectin [29,30].

Overlapping bands in the range of 2800–3000 cm^−1^ occur in the case of stretching vibrations of C-H bonds. At the wave number of 1630 cm^−1^, an intense and narrow band of amide I is observed, which corresponds to the stretching vibrations of C=O bonds and N-H bonds. The band at 1544 cm^−1^ comes from Amide II, which consists of stretching vibrations of C-N bonds and deformation vibrations in the plane of N-H bonds [31]. The absorbance signal peaking at 1400 cm^−1^ occurs when the incident radiation induced an in-plane deformation vibration for the C-OH and CH_2_ groups. The out-of-plane deformation vibration of the methylene group can be assigned to the 1335 cm^−1^ band. The maximum at 1240 cm^−1^ is the Amide III band of the deformation vibrations of the N-H group and the stretching vibrations of the C-N group, which are influenced by the δ(C-O) and ν(C-C) vibrations. Stretching vibrations of the C-O group present, for example, in a glycerol molecule give an intense band for the wave number at approximately 1040 cm^−1^, while for wave numbers: 900 cm^−1^–840 cm^−1^, these are signals from the groups: C-O and C-O-SO4, respectively. The spectra for gelatine mixtures with the addition of agar in various weight fractions are shown in Figure 3 and Figure 1.

The spectra obtained during measurements for blends containing agar contain groups originating from gelatine and glycerine in their structure, which give characteristic bands in the spectrum of each sample. In the spectrum of A35 and A50 composites for the range of 2800–3000 cm^−1^, the absorbance of two bands corresponding to the stretching of C–H bonds (2910 cm^−1^ and 2840 cm^−1^) increased, while the Amide A band is less intense. The absorbance for a blend with the addition of agar in 50 parts. wt. (A50) increased in the range from 1350–1150 cm^−1^ and 800 to 1000 cm^−1^. The higher intensity of the band is probably caused by the presence of sulfide groups S=O and glycosidic bonds of the polysaccharide. The increase in absorbance indicates the presence of a greater number of connections, which may translate into the mechanical and functional properties of the composites. Gelatine and polysaccharides may interact through different mechanisms due to their chemical properties. One mechanism is the electrostatic attraction between the positively charged amino groups in gelatine and the negatively charged carboxylate groups in polysaccharides. Another mechanism is hydrogen bonds between the hydroxyl groups of polysaccharides and amide groups in gelatine. Furthermore, hydrophobic interactions between gelatine and polysaccharides can also occur where nonpolar regions interact with each other.

Figure 4 shows a comparative spectrum for gelatine samples with the addition of different weight percentages of carrageenan. As in the case of samples with agar added, most of the peaks appearing in the spectrum come from gelatine and glycerine. As the content of carrageenan in the biopolymer structure increases, the intensity of the peaks increases, which may affect the functional properties of the composites due to the larger number of galactose and anhydrogalactose groups. Typical bands for carrageenan are signals in the ranges of 1350–1150 cm^−1^ and 900 cm^−1^ and 840 cm^−1^, corresponding to the groups of sulfide groups S=O, 3,6-anhydrogalactose, and galactose 4-sulfate. Figure 2 shows the possible interactions between the carrageenan and gelatine blend.

Carrageenan is a sulfated polysaccharide containing glycosidic bonds between the monomer units. Gelatine, on the other hand, is a protein that contains peptide bonds between amino acid residues. These two molecules can interact with each other through a variety of mechanisms, including electrostatic attraction, hydrogen bonds, and hydrophobic interactions.

The glycosidic bonds of carrageenan can interact with the peptide bonds of gelatine through hydrogen bonds. Hydrogen bonding occurs between the hydroxyl groups of carrageenan monomers and the amide groups of gelatine amino acids. This interaction can lead to the formation of a complex between the two molecules, which can have different properties depending on the specific type of carrageenan and gelatine used.

### 2.3. Nuclear Magnetic Resonance Spectroscopy NMR

The proton spectrum of gelatine consists of characteristic signals originating from amino acids such as glycine, proline, threonine, leucine, valine, arginine, hydroxyproline, tyrosine, and phenylalanine (Figure 5, Table 3). The assignments of the signals are consistent with the literature data [32,33]. Table 3 presents the assignments of individual signals.

The spectrum of glycerine consisted of three characteristic signals (Figure 6). One originating from the CH group with a shift of 3.78 ppm, and two from the CH_2_ groups with shifts for the proton at 3.56 and 3.67 ppm (Table 4). Figure 7 represents the spectrum of gel. As can be seen, it shows characteristic signals which were observed for both gelatine and glycerine.

Carrageenans are a group of galactans that are soluble in water and contain sulfate groups. They consist of a chain of linked 3-linked β-d-galactopyranose (G-units) and either 4-linked α- d-galactopyranose (D-units) or 4-linked 3,6-anhydro-α-d-galactopyranose (DA-units). These units repeat to form the structure of carrageenans. The ^1^H NMR spectra of carrageenans present typical deshielded signals corresponding to the anomeric hydrogens at 5.24 and 5.38 ppm (Figure 8). The carbon spectra consist of 12 signals, corresponding to the 12 carbon atoms of the repeating disaccharide unit (Figure 9). The spectra are consistent with the literature data [34,35].

Unfortunately, the ^1^H spectrum of the sample CGN15 presented in Figure 10 only shows intense signals from glycerine which makes the signals from amino acids in gelatine invisible.

The main sugar component of agar is agarose, which represents about 70%. Agarose is a polysaccharide that is linear and mostly free of sulphate. Its sugar structure is made up of alternating β-d-galactopyranose units linked at the 1,3 position and 3,6 anhydro-α-l-galactopyranose units linked at the 1,4 position. This polysaccharide is well-known for its ability to form gels and is commonly utilized for this purpose. Figure 11 shows a representation of the repeated unit in agarose. In Table 5, G is used for the anhydrogalactose unit, G′ for the D-galactose unit, and G″ for the substituted D-galactose unit [36,37].

The assignments of the signals are consistent with the literature data [38,39]. Figure 12 and Figure 13 present the proton and carbon spectra for agar, respectively, and Figure 14 shows the proton spectrum for the biopolymer blend containing the addition of agar.

Figure 14 represents the spectrum of A15. As can be seen, it shows characteristic signals which were observed for both the agar and gel gelatin–glycerine. In the collective spectrum (Figure 15) showing signals from agar 1, gelatin–glycerine gel 2, and the gelatine blend containing agar 3, one can observe signals probably coming from the amino acid and polysaccharide units. The gelatine mixture is a mixture of amino acids, equipped with carboxyl and amino groups that form peptide bonds present in key structures such as hydroxyproline, proline, and glycine, but also containing aromatic groups in their structure, such as phenylalanine or tyrosine. Generally, in organic compounds, electron-withdrawing substituents, such as Ar-H, R-(C=O)-NH, Ar-OH, and SO_3_-, located in the vicinity of a given atom cause a shift towards greater exposure, i.e., towards a higher ppm value. The introduction of agar into the gelatine–glycerine structure did not reveal these signals due to the large biomolecule that is gelatine, which, in turn, results from the fact that T2 relaxation times are very short, and, therefore, in a mixture where there may be low-molecular-weight compounds and polymers, the latter are lost in the baseline [40].

### 2.4. Determination of Water Absorption

Determining the absorption curve of the samples allows for the estimation of the most favorable composition of the composites from the point of view of the functional properties. For this purpose, the mass of the samples is measured as a function of time until a plateau point is reached. The percentage change in the mass of the composites is shown in Figure 16.

Water absorption is a measure of the amount of water that a polymer blend can absorb. This is important because water can affect the mechanical properties of the mixture, such as its strength and stiffness. In addition, water can also cause the polymer mixture to swell, which may lead to a reduction in its dimensional stability. Therefore, it is important to understand the water absorption properties of the polymer blend to ensure it performs optimally in various environments [41,42].

Based on the water absorption curve, it can be concluded that the reference sample (Control) absorbed the most water compared to the other samples—the mass ultimately increased by 93%, and the sample swelled noticeably after the first 10 minutes of analysis. Hydroxyl groups in the pyranose ring are responsible for such absorption, which initiates the formation of hydrogen bonds with water. 

A larger addition of agar resulted in a smaller change in mass when immersing the biopolymer blends in water. A larger amount of polysaccharide allowed for the creation of more stable interactions between biopolymers, which is confirmed by an FTIR analysis [43].

### 2.5. Density of the Blends 

The results of density measurements presented in Table 5 indicate that the sample without the addition of polysaccharide has the lowest hydrostatic density and is similar to the value of the composition with the addition of agar in 15 parts weight (A15). Blends with the addition of polysaccharides increase in density as the amount of this substance increases. The density depends on the chemical composition and structure of the polymer—in this case, on the content of additional ingredients, i.e., the structure of the carrageenan or agar.

The chemical composition of a polymer affects its density through the mass of the atoms it is made of. For example, polymers containing atoms with a high atomic weight will have a higher density than polymers containing atoms with a lower atomic weight.

The structure of a polymer affects its density by the way the atoms are connected to each other. For example, linear polymers will have a lower density than branched polymers. This is because linear polymers have a smaller volume per unit mass.

Gelatine is a polypeptide that consists of amino acid residues linked together. Carrageenan and agar are polysaccharides that consist of sugar molecules linked together. Due to its chemical composition, powdered gelatine has a relatively low density of approximately 1.3 g/cm^3^ [32]. Powdered carrageenan [33] and agar [32] have a higher density of approximately 1.5 g/cm^3^ and 1.8 g/cm^3^, respectively. The structure of gelatine is linear. Carrageenan and agar have a branched structure [44]. Therefore, gelatine has a lower density than carrageenan and agar. The results of the hydrostatic density of the made biopolymer blends are presented in Table 6.

### 2.6. Contact Angle and Surface Free Energy SFE

Surface energy is a measure of the attractive force between molecules located on the surface of the material. This is an important feature of materials because it affects their physical and chemical properties. Determining the surface energy using three liquids, water, ethylene glycol, and diiodomethane, required measuring the contact angles of drops of each liquid on the surface of the samples. The measurement results in the form of average surface energy values are presented in Table 7 and correspond to the literature results [42,45].

The polar component for samples with added polysaccharides decreases with the increasing weight of these additives, which indicates a decrease in the hydrophilic properties of the blends made. These data correspond to the results of the analysis of the absorption curve. This is probably related to the formation of macromolecular connections as a result of covalent interactions resulting from the interactions of the -OH moiety derived from the polysaccharides. This reduces the interactions of hydroxyl groups on the composite surface, which is confirmed by an FTIR analysis (Figure 3 and Figure 4). However, the polar component increases with the increase in the weight fraction of additives. This is related to the increase in the number of hydrophobic groups on the sample surface.

The surface free energy of blends with the addition of polysaccharides is higher than the SFE of the control sample and increases with the increase in these additions in the blends made. The higher SFE value indicates a more favorable cross-linking of samples with polysaccharides; the lower number of polar groups near the surface and on it results in the reduced penetration of polar solvents like water.

### 2.7. Mechanical Properties

In order to determine the mechanical properties of the samples, the hardness, tensile strength (T_S_, MPa), and elongation at break (E_b_, %) were determined. The hardness of the samples was determined using a Shore type A hardness tester, and a Zwick Roell testing machine was used to obtain the values of the tensile strength and relative elongation at break. Relative elongation is the ratio of the final length of the sample to its initial length. This is a measure of the material’s elasticity. Tensile strength is the maximum force that a material can withstand without breaking. This is a measure of the material’s strength. The results and their standard deviations are presented in Table 8.

Based on the results obtained, it can be concluded that the introduction of polysaccharides into the blends leads to an increase in hardness and a decrease in elongation at break, and the results are comparable to the results of measurements performed for similar systems [42,44,45,46,47]. The decrease in the value of the relative elongation of compositions containing polysaccharide may be related to the formation of a larger number of gelatine–polysaccharide bonds in the blend network, formed due to the presence of functional groups attached to the pyranose ring of agar and carrageenan. The tensile strength increases for samples with the addition of agar. This is probably related to the formation of covalent bonds between gelatine and agar. Carrageenan composites were characterized by a relatively low tensile strength, lower than the control sample, and an additionally higher proportion of carrageenan in the gelatine matrix reduced the tensile strength of the composite, which would indicate the formation of weak bonds such as hydrogen bonds in the network or the blocking of the active groups of carrageenan and gelatine during synthesis. Agar contains two units of SO_3_ sulfonating groups in two furanose rings, while carrageenan contains only one. This difference in sulfonated groups between the two substances has important implications for the formation of covalent interactions in the structure of the mixture. In particular, the presence of more sulfonated groups in the agar may lead to the formation of more covalent interactions in the mixture structure. This, in turn, may result in more favorable T_S_ (tensile strength) parameters of the mixture.

Gelatine composites are characterized by a relatively high relative elongation, usually ranging from 41.7 to 230%. This is higher than other composite materials such as polyester composites (5 to 20%) or polyurethane composites (10 to 30%).

The presented gelatine blends are characterized by a relatively low tensile strength compared to the strength of the other composite materials. For example, gelatine composites with the addition of glass fibers can have tensile strengths ranging from 20 to 50 MPa.

Gelatine blends are characterized by the Shore hardness, which is similar to the hardness of other composite materials, such as gelatine composites with the addition of glass fibers, which have a Shore hardness of 50 to 70.

### 2.8. Thermal Analysis

In order to determine the thermal properties of the prepared samples, differential scanning calorimetry (DSC), thermal gravimetric analysis (TGA), and softening temperature using the Vicat method were carried out. Figure 17 shows thermograms for the made blends.

The thermogravimetric method was used to determine the thermal stability of blends made of gelatine, agar, and carrageenan. Biopolymer blends with the addition of polysaccharides have a TGA curve analogous to the standard in terms of the occurrence of transformations [46]. The addition of agar and carrageenan caused the first mass loss to shift to higher temperatures, which is related to the more strongly bound water in the matrix through the water–polysaccharide hydrogen interactions. In the case of carrageenan, the first weight loss moved to the range of 180–220 °C for all composites. Blends with the addition of A15 and A30 agar required higher temperatures for the degradation of gelatine, glycerol, and agar—220–290 °C and 290–400 °C. The transformations that occur in carrageenan blends are identical to those in the agar addition. A higher content of carrageenan did not cause any shifts in mass losses in the temperature range of 220–290 °C and 290–400 °C. In the range of 220–290 °C, the CGN35 composite lost less mass, and, in the range of 290–400 °C, all gelatine–polysaccharide samples lost less mass.

Table 9 shows the results for a loss of 5% and 50% of the mass of the tested samples and residues after thermal decomposition. The temperatures for a 5% mass loss had similar values, because the mass loss occurred due to the loss of water bound in the polymer matrix. In the case of all carrageenan blends, the temperature needed to decompose 50% of the sample mass increased by approximately 20 °C. The residue after the charring of the samples with polysaccharide increased due to the higher ash content of polysaccharides compared to gelatine.

All thermal transformations that occurred under the influence of the temperature were endothermic transformations. Gelatine composites give three phase transition temperature readings, which are described as the glass transition temperature Tg and the melting temperature Tm. In the literature, the glass transition temperature of gelatine varies depending on its origin or different extraction methods. The melting point of gelatine (Tmelt) can vary depending on the histological origin of the animal tissue from which it was extracted, e.g., pork gelatine 38–42 °C, bovine gelatine 30–35 °C, fish gelatine 22–28 °C, and poultry gelatine 25–30 °C [4].

The results for differential scanning calorimetry (DSC) for the gelatine–agar and gelatine–carrageenan matrices are presented in Table 10 below.

The glass transition temperatures Tg for the standard and the blend with the addition of agar have similar values, but, with the increase in the weight share of agar in the blend, an increase in the heat capacity can be observed. Heat capacity is the amount of heat needed to raise the temperature of a material by one degree Celsius. An increasing heat capacity means that the material requires more energy to heat it.

The addition of carrageenan to the polymer matrix resulted in an increase in the glass transition temperature Tg and an increase in the heat capacity Cp, which proves that carrageenan increased the thermal stability of the gelatine matrices. The blend with the highest weight share of CGN50 carrageenan turned out to be the most stable, and its heat capacity was 2.736 J/g∙K.

The effect of polysaccharides on the increase in the glass transition temperature of gelatine is associated with the formation of non-covalent or covalent complexes that can change the molecular structure and physical properties of gelatine, leading to an increase in its glass transition temperature. The thermal properties of gelatine biopolymers are similar to those of other composite materials, such as polyester or polyurethane composites. The glass transition temperature of gelatine blends reaches up to 110 °C. This temperature is close to the Tg of some other materials, such as polycarbonate of 147 °C and polyethylene terephthalate (PET) of 80 °C. The increased heat capacity observed in gelatine mixtures after the introduction of polysaccharides can be attributed to the formation of a stable network between the protein and carbohydrate components, resulting in an improved thermal and mechanical stability. Comparing the results regarding the heat capacity of biopolymer blends with the properties of other polymers, it becomes obvious that this value closely coincides with the range of heat capacity of polyesters (from 1.3 to 1.7 J/g∙K) and polyurethanes (from 1.5 to 2.0 J/g∙K).

To determine the softening point, a Vicat analysis was performed for various biopolymer blends. The softening temperature Tm indicates the transition of the biopolymer into a plastic form in a gradual manner. The control sample (without added polysaccharides) had the lowest measured temperature value (47.43 ± 1.09 °C). The addition of agar increases the softening temperature Tm, but this change was much smaller compared to the addition of carrageenan. The highest softening temperature was achieved by the CGN50 composite, which was approximately four times higher than the temperature of the Control composite and amounted to 192.25 ± 0.85 °C. The influence of polysaccharides on the increase in the softening temperature of gelatine mixtures is related to their developed structure, availability of functional groups, and the introduction of polysaccharide derivatives.

### 2.9. Biodegradability and Thermo-Oxidative Aging of Samples

#### 2.9.1. Thermo-Oxidative Aging

In order to determine the resistance of the samples to thermo-oxidative aging, accelerated aging measurements were performed. The samples were placed in a thermo-oxidation chamber for 7 days. Samples subjected to the thermo-oxidation process were subjected to various analyses to determine the impact of aging on the physico-chemical properties of polymer blends.

#### 2.9.2. FTIR Spectroscopy after Thermo-Oxidative Aging

Thermo-oxidative aging of samples affects their structure, and these changes can be observed using infrared spectroscopy. The difference is the decrease in absorbance after thermo-oxidative aging for all compositions. After thermo-oxidative aging, agar blends absorbed less radiation compared to composites with carrageenan. The conditions to which the samples were subjected had a greater impact on the agar matrices. For none of the composites after thermo-oxidative aging, no change in the position of the bands is observed, so the same vibrations still occur, only with a lower efficiency.

Figure 18 and Figure 19 show the spectra of A50 and CGN50 composites before and after thermo-oxidative aging. For the agar composite, a band of higher intensity occurs for the stretching of the N-H bond in the range of 2800–3000 cm^−1^. The increase in the number of amino groups may be caused by cross-linking reactions that occur during thermo-oxidative aging.

A decrease in the intensity of the C-O bond stretching band in the range of 1000–1200 cm^−1^ may indicate a reduction in the number of carbonyl groups in the matrix. The reduction in the number of carbonyl groups may be caused by oxidation reactions that occur during thermo-oxidative aging [48].

#### 2.9.3. Photos with an Optical Microscope

The microscope was used to take magnified photos of the topography of the composites before and after aging to compare the surface controls. The photos are summarized in Figure 20. Based on the images taken, it can be concluded that all samples contained air bubbles, craters, and salt crystals, which are most likely the result of the uneven evaporation of water from the protein matrix occurring during processing.

The Control composite was the most homogeneous and contained fewer agglomerates than A15 and K50, and the structure had no distortions. In the photos of blends subjected to thermo-oxidative aging, you can see scratches on the sample surface resulting from the effect of the temperature on the sample during processing.

The introduction of agar into the gelatine matrix causes distortions on the surface and these changes are visible for the A15 composite, and this effect deepens with an increase in the weight fraction of the polysaccharide additive. Micropores were observed on the topography of each sample, the number of which decreased with the increasing polysaccharide content.

In the photos of composites with carrageenan, convex structures are observed, changing their shape depending on the amount of carrageenan added to the gelatine matrix. CGN15 is characterized by the presence of a large number of elongated, irregularly arranged carrageenan agglomerates. In the remaining samples with the addition of carrageenan (CGN35 and CGN50), there are fewer agglomerates, but they are larger and take a spheroidal shape.

Composites subjected to thermo-oxidative aging are characterized by the presence of distortions (mainly A15), cracks (mainly A35), and an increase in the number of agglomerates (CGN50). The surface of the carrageenan composites, compared to the agar composites, did not change significantly, but the number of agglomerates increased after aging, which was most likely influenced by the content of salt precipitated in the form of crystals. The longitudinal, convex agglomerations (CGN15) visibly increased in volume after aging, and a similar effect was caused by the aging of the CGN35 and CGN50 composite [49].

#### 2.9.4. Determination of Color Change after Aging

The color change was measured in the CIELAB space using a UV–Vis spectrophotometer. The samples before aging were used as standard samples. The results of the color change parameter dE*ab and color changes on the surface of samples are presented in Table 11. The dE*ab parameter describes the differences in color compared with the samples before aging, also tested for other gelatine samples [42].

In the case of agar, the total color change was above the value of 5, so the change in the color of the sample after thermo-oxidative aging is noticeable to the human eye. The higher the weight of the polysaccharide in the blend, the smaller the difference between the colors of the samples. The color change of carrageenan compositions due to aging will not be noticeable to a standard observer, which was confirmed by a total color difference value below unity. Carrageenan gives the gel matrix a greater resistance to color change compared to agar (ΔE 0.42–0.89).

#### 2.9.5. Determination of Changes in Mechanical Properties after Aging Processes

Figure 21 presents the results of the hardness analysis in Shore’s scale A for the blends before and after thermo-oxidative (TO) aging.

Figure 21 shows the change in Shore A hardness for gelatine mixtures under the influence of the temperature. From the presented results, it can be concluded that thermo-oxidative aging increased the hardness by an average of two times. The greatest increase was observed for samples with the addition of carrageenan, from 168% to 119%. This may be due to the formation of oxygen radicals, which leads to the secondary cross-linking of the matrix during thermo-oxidative aging. The formation of a secondary network significantly affects the mechanical properties of materials.

#### 2.9.6. Determination of the Change in SFE after Thermo-Oxidative Aging

Determining the surface free energy SFE using three liquids, water, ethylene glycol, and diiodomethane, required measuring the contact angles of drops on the sample surface, and then calculating the surface energy values after thermo-oxidative aging.

The SFE surface free energy for samples with added polysaccharides was higher than the surface free energy of the control sample (Figure 22). Thermo-oxidative aging resulted in an increase in surface free energy for all blends. The increase in surface energy is probably related to the fact that carbonyl groups forming bonds during thermo-oxidative aging cross-link the structure inside.

The contact angles for the polar liquid (distilled water) increased for all mixtures after aging and ranged from 114° to 135°. This indicates a change in structure and the formation of more hydrogen bonds due to the presence of free radicals in the matrix. The largest change in the contact angle was observed for the sample with the addition of agarose, reaching 123.15°. However, for a non-polar liquid (diiodomethane), there was a decrease in contact angles for all mixtures, which may indicate a decrease in non-polar groups (for example, amine). The same relationships are reflected in the results of surface energy, which increases with thermo-oxidative aging, which may confirm secondary cross-linking in the biopolymer matrix.

Thermo-oxidative aging refers to the degradation of a material by exposure to heat and oxygen, resulting in the formation of oxidation products on the surface. Thermo-oxidative aging can cause the breakdown of chemical bonds in the material, which leads to the formation of new functional groups and the rearrangement of molecular structures. This may result in an increase in surface energy due to the presence of more polar and reactive groups on the surface, as well as a more irregular and less ordered surface structure [45].

#### 2.9.7. Microbial Aging

In order to determine the resistance of the samples to composting, a test was carried out in accordance with the PN-EN ISO 846 standard [42,50]. Universal soil with a pH of 6.0–7.0 and paddle-shaped samples with the shape described in Table 1 below were used. Blends of each type were placed on a layer of soil in a container, then covered with another layer of soil, and then the containers were placed in a climatic chamber for 15 days at a temperature of 30 °C and a humidity of 80%. After the measurement, the complete degradation of the samples was observed.

In order to influence the composting on the degradation of gelatine blends, an analysis of the mass loss of samples was carried out to determine the microbiological aging of gelatine mixtures. The results are shown in Figure 23.

The figure shows the weight loss of mixtures under the composting, which behave similarly to mixtures described in the literature [42]. Composites subjected to composting in the first stage of biological decomposition increased their volume, except for the Control and A15 samples, in which a percentage decrease in mass was observed after 24 h. After 48 h, a mold colony was observed on the control mixture and the mixture with the addition of A15 agar. The control sample biodegraded after 144 h. The introduction of agar into the blends resulted in an increase in the composting time from 168 h to 288 h along with an increase in the weight of agar in the blend. Increasing the amount of carrageenan in the samples accelerates the decomposition of the biological blend.

The decomposition of gelatinous blends occurs as a result of the sorptive action of the soil environment and the soil micro-organisms contained therein, during hydrolytic and enzymatic degradation mechanisms, in which micro-organisms are precisely involved. The enzymes produced by micro-organisms have the ability to break down gelatine and polysaccharides found in samples, transforming them into smaller fragments that can be easily metabolized by these same micro-organisms. In the case of gelatine, enzymes break down the covalent and hydrogen bonds in collagen molecules. In the case of polysaccharides, enzymes break down the glycosidic bonds between sugar residues. In the case of samples modified with polysaccharides, the structure is more complex than that of the Control sample. The complex structure of blends makes it difficult for enzymes equipped with both polypeptide and polysaccharide apparatuses to access its molecules, which slows down the degradation process.

## 3. Materials and Methods

### 3.1. Materials

Polymer biocomposites were prepared for testing, the ingredients of which were: gelatin with a Bloom index of 200° (pH 5–8, Mw 30 kDa, FoodCare Sp. z o. o. Zabierzów, Poland) and glycerin (pH 6–8, d = 1.26 g/ cm^3^, Mw 92 kDa, Flash point 160 °C, Chempur, Piekary Śląskie, Poland) added to each composition in the same amounts. Natural polysaccharide additives, i.e., modifying substances, were agar E406 (pH 1.5% solution 6.0–7.5; gelation temperature 33–36 °C, liquefaction temperature above 86 °C, gelling power 850 g/cm^3^) and carrageenan E407 obtained from seaweed belonging to the class Rodophyceae (AGNEX, water solubility 50 g/L at 20 °C, pH 7–10, Costa del Sol, Spain). Sodium base was used (pH 13–14, freezing point 12 °C, Mw 22 mPas at 40 °C, Eurochem BDG Sp. z o. o. Tarnów, Poland).

### 3.2. Methods

#### 3.2.1. Fourier-Transform Infrared Spectroscopy

Infrared Fourier spectroscopy analysis of bioplaics samples in order to study their structure was performed with the A Nicolet 6700 spectrophotometer (Thermo Scientific, Waltham, MA, USA). It should be noted that the spectroscopy analysis was carried out at a resolution of 0.25 cm^−1^ and in the wavelength range from 4000 to 400 cm^−1^. In the FTIR analysis, the ATR method, i.e., attenuated total reflection of IR radiation, was used. After synthesis, the polymer blends were formed under pressure of 10 MPa and temperature of 140–150 °C into plates with a thickness of 1 mm.

#### 3.2.2. Nuclear Magnetic Resonance (NMR)

All spectra were recorded with a JEOL spectrometer operating at a ^1^H frequency of 400 MHz. The instrument was equipped with 5 mm HFX-Royal probe. All experiments were performed at 343 K. The standard proton spectra were acquired with a calibrated 90° pulse for 16 scans collecting 16 K data points over a spectral width of 12 ppm. The TSP peak at 0 ppm was used as a chemical shift standard for ^1^H spectra.

#### 3.2.3. Determination of Water Absorption

Water absorption was measured by testing the absorption and drying of the blends produced. It consisted of two stages: the first was to weigh the samples before placing them in distilled water, then put them in distilled water and weigh them again after 10, 20, 30, 40, and 60 min, according to PN-ISO 8361-1:1994 [51]. In this way, the dependence of the change in mass as a function of time during absorption was obtained. When the swelling sample stopped changing mass, the second stage of measurements, drying, began. Similarly, the samples were weighed, at the same time intervals, but temperature was the factor acting on the sample after it was placed in a thermal chamber (GmbH, Tuttlingen, Germany), set at 100 °C. The weights measured at the intervals made it possible to determine the dependence of the change in weight as a function of time during drying.

#### 3.2.4. Density of the Blends

The density of the composites was measured using a hydrostatic method, taking three measurements for three samples—a balance reading in air and in an immersion liquid. The system consisted of an electronic balance with a stand and beaker, as well as a rack for two connected pans and a thermometer, which was used to read the temperature relating directly to the density of the liquid used. First, the sample was weighed on the pan not immersed in the immersion liquid, obtaining the mass in air, then on the lower pan—the mass in the immersion liquid. The measurement was carried out at a constant temperature; the samples were adequately degreased and of an appropriate size, which gave a comparable result for a series of composites.

#### 3.2.5. Surface Morphology

The polymer blends were examined using a LAB 40 Series metallographic microscope (OPTA-TECH, Warsaw, Poland), which made it possible to compare the topography of the samples before and after thermo-oxidative aging at magnifications of ×50, ×100, ×200, and ×500. The thickness of each sample was in the range of 3–4 mm. Using 200× magnification was optimal for comparing blends in terms of structure, as it allowed us to observe the most differences between the tested materials. 

#### 3.2.6. Mechanical Properties

The mechanical properties of bioplastics were tested using two devices: Zwick 1435 (Zwick/Roell, Radeberg, Germany) and a digital Shore type A hardness tester (Zwick/Roell, Herefordshire, UK). Zwick 1435 was used to test breaking strength (MPa) and elongation at break (%). The hardness tester was designed to test the Shore hardness (°Sh) of bioplastic samples. To perform these tests on the Zwick 1435, test specimens were first cut from material 1.6–1.8 mm thick and 150 mm long. The measurement speed during the analysis of breaking strength and elongation at break on the Zwick 1435 machine was 50 mm/min, while the preload was 0.1 N. 

For the analysis of Shore hardness, bioplastic samples were prepared in such a way that they had a thickness of about 4 mm. The tests were carried out on a device with a pressure force of 12.5 N and an indenter with a hardness of 35° Sh. Hardness analysis with a digital device consists in examining the difference between the initial depth and the depth of the indentation in time. The test consisted in placing the prepared sample on the hardness tester’s stand, and then lowering the needle onto the sample, along with reading the hardness result. This process is repeated five times at different locations on the sample. The final result is the average of the read data with the standard error.

#### 3.2.7. Thermal Properties

The measurement of thermal properties of bioplastics was carried out using thermogravimetric analysis (TGA) and differential scanning calorimetry (DSC) together with the computer program Mettler Toledo TGA/DSC (Mettlet Toledo, Greifensee, Switzerland) with calibration with an indium or zinc standard. For testing, samples weighing 4–10 mg were placed in open aluminum crucibles with a volume of 100 µL, and then the crucibles were placed in a measuring device that heated the samples from 0 to 200 °C at a rate of 10 °C/min in an argon atmosphere. At one point in the test, the gas was changed from argon to air and the flow rate was 50 mL/min.

The determination of the Vicat softening temperature T_m_ for amorphous or crystalline-amorphous materials was carried out by finding the temperature at which the indenter penetrated 1 mm into the sample. Samples were prepared in a square shape and thicknesses ranging from 3 to 6.5 mm, and then tested using a softening temperature measuring apparatus according to Vicata D-Vicat.HDT/3/300FA (IDM Instruments, Melbourne, Australia). The liquid of the apparatus, properly stirred, was heated at a rate of 90 °C/h, while the pressing force was 0 N.

#### 3.2.8. Surface Free Energy and the Degree of Cross-Linking

For further morphological analysis of bioplastics, the OCA 15EC goniometer (Dataphysics, Filderstadt, Germany) was used to perform the surface free energy (SFE) study. The method used to test SFE is the Owens, Wendt, Rabel, and Kaelble method (OWRK). In this method, three measuring liquids are used, polar (distilled water), non-polar (diiodomethane), and boundary (ethylene glycol), to determine the contact angles of the liquid on the sample. Then, to calculate the surface free energy (SFE), Equation (1) was used, in which *γS_p_* is the value of the polar component, and *γS_d_* is the value of the dispersion component (mJ/m^2^):(1)$$\gamma S={\gamma S}_{p}+{\gamma S}_{d}$$

#### 3.2.9. Thermo-Oxidative Aging

To analyze the biodegradability of samples, they were prepared in the shapes of paddles and placed in a thermal chamber for 7 days at 70 °C according to ISO 11358 [52].

#### 3.2.10. UV–Vis Spectroscopy and Color Stability

The CIE-Lab UV–Vis spectrophotometer (UV-VIS CM-3600d, Konica Minolta Co., Chiyoda, Tokio, Japan) was used to analyze the color change after aging and the color difference ∆E was determined in accordance with the PN-EN 105-J01:2002 [42] and PN-EN-105-J03:2009 standards [42]. The advantage of the CIE-Lab system is the ease of comparing colors, which is important in the production of products with complex color patterns. The CIE-Lab color space expresses the following color features, lightness, saturation, and hue, which means that the color analysis of the sample is described in the CIE-Lab space in the three-co-ordinate system: L, a, and b, where “L” is the brightness parameter (maximum value 100, representing a perfectly reflective diffuser; minimum value of zero, representing black), “a” is the red to green spectrum axis, and “b” is the yellow to blue color axis, and the “a” and “b” axes have no numerical limits. The color change, dE*ab, was calculated according to the following equation, Equation (2):(2)$$dE\times ab=\sqrt{\left(∆a^{2}\right)+\left(∆b^{2}\right)+\left(∆L^{2}\right)}$$

#### 3.2.11. Soil Biodegradation

The compostability of bioplastic samples was carried out in order to confirm the biodegradability of the material in soil conditions. In accordance with the PN-EN-ISO 846 standard, the samples were placed in universal soil for 14 days at 30 °C and 80% air (HPP108 Memmert climatic chamber, Memmert GmbH + Co.KG, Schwabach, Germany).

## 4. Conclusions

The use of non-degradable plastics has led to a significant increase in plastic waste, which is a serious environmental problem. Biodegradable biopolymers and their derivatives have been proposed as a potential solution to this problem. In order to obtain a stable biopolymer coating, gelatine mixtures modified with polysaccharides such as agar and kappa-carrageenan were prepared and their parameters were determined using FTIR, TGA/DSC techniques, contact angle measurements, and surface energy characteristics.

Polymer blends containing a higher weight share of polysaccharide–agar achieve an increase in the degree of cross-linking compared to reference samples, which is associated with a change in specific density and surface energy. The increase in surface energy may be due to the presence of a polar polysaccharide component. The factors contributing to this behavior also include, in addition to a larger number of polar groups, e.g., hydroxyl (-OH) and carboxyl (-COOH) groups, the formation of aggregates. Fragments of macromolecules having the above groups in their structure and forming a three-dimensional network with the host polymer molecules may be active in forming hydrogen bonds with the host polymer molecules, which leads to the strengthening of the composite structure and, consequently, an increase in the softening temperature. Carrageenan entering the network with the parent polymer molecules acts as a scaffolding, which additionally strengthens the structure of the composite and makes it difficult to soften.

The degradation progress in the process of composting and thermo-oxidative aging results in an increase in the hardness of the obtained systems, with the greatest increase observed in the case of samples with carrageenan. The surface free energy of the mixtures also increased, probably due to the formation of polar groups cross-linking the structure.

The composting test resulted in the complete degradation of the samples, the introduction of agar extended the composting time, and increasing the carrageenan content in the gelatine mixture accelerated the decomposition of the sample subjected to the composting process. The enzymatic degradation of the mixtures occurs through the breakdown of covalent and hydrogen bonds in collagen molecules and glycosidic bonds between sugar residues.

However, there is still a need for further research to improve the physicochemical parameters of the presented biopolymer mixtures. It is important to conduct more detailed and in-depth studies (e.g., electron microscopy (SEM/TEM) or X-ray diffraction) to better understand the physical and chemical properties of the described mixtures, which may, ultimately, lead to the development of more efficient and effective products. Indeed, in order to improve the mechanical properties, the addition of cross-linking compounds, e.g., glutaraldehyde, calcium ions, and chitosan, can be used. Moreover, the introduction of thermal stabilizers, e.g., antioxidants, can improve thermal properties.

## Data Availability

The data are contained within the article.

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
