# Peer review of "Gelatine Blends Modified with Polysaccharides: A Potential Alternative to Non-Degradable Plastics"

_ijms, 2024, doi:10.3390/ijms25084333_

Round 1
Reviewer 1 Report
Comments and Suggestions for Authors
As selected biopolymers are degradable. What is the need for checking microbial aging and soil biodegradable tests?

The introduction parts need to be improved. How long the stability of film?
Author Response
Thank you for your comments. We have made corrections in accordance with the guidelines.
- The work concerned the production of ecological polymer blends, on the basis of which elements or entire products with forms and shapes, depending on the designed processing method, can be produced, with end uses as materials, among others, in packaging applications.
- Yes, polymer blend mixtures were synthesized using a chemical reactor equipped with a cooler, a mechanical stirrer, a temperature controller and a heating jacket. After the synthesis was completed, the whole was transferred to a vacuum evaporator to remove the remaining solvent. An efficiency of 85% was achieved in the process. The unformed material was subjected to a processing process of pressing at a temperature of 140-150°C and under a pressure of 10 MPa in order to give the desired shape - in the case of the research, these were plates with dimensions 100mm x 100.mm x 1.mm. Subsequently, an element dimensioned for individual examinations and tests was cut from the boards using an automatic die.
- In addition to producing polymer blends, our intention was to conduct a number of physico-chemical tests verifying the properties of polymer blends. The introduction of polysaccharides strengthened the structure of the polypeptide matrix, stabilizing the polymer network using mainly hydrogen bonds and ionic interactions.
- Indeed, the importance and emphasis on the method of producing polymer blends is extremely important. We have made corrections regarding the specification of the synthesis method and processing methods of refined polymer materials.
- The reaction efficiency has been supplemented in the "Preparation of the Blends" section, which is extremely important due to the assessment of the effectiveness of the chemical process. the estimated efficiency is 85%, which means that for every 100 grams of moles, a milligram etc. introduced substrates, there were 85 grams of moles, milligrams, etc. a truly refined product.
- Literature references were introduced in the discussion of results section to validate the obtained results.
- Indeed, it would be possible to tabulate the obtained results versus the results from literature reports, but due to the extensiveness of the article, i.e. the number of figures and tables, the introduction of an additional table would increase the volume of the article, therefore only new literature reports were added in the introduction part.
- Figure 1 and diagrams 1 and 2 have been changed.
- The drawings included in the NMR analysis, their description and methodology were prepared by an independent laboratory of the Institute of Organic Chemistry of the Lodz University of Technology - not located in our parent unit. Therefore, we did not interfere with their quality and presentation.
Why are polymer blends tested using depth tests and what is the stability of the films?
The tests are designed to confirm the rate of degradation after a period of use of the presented materials. Blends in the form of cross-linked films do not degrade over a long period of time (even up to 2 years, as confirmed by our previous studies), but only when introduced into controlled environmental conditions do they begin to degrade. In soil decomposition occurs even up to 14 days.

Reviewer 2 Report
Comments and Suggestions for Authors
The problem of creating biodegradable plastics, as an alternative to mass-produced polyethylene, polypropylene, etc., is extremely relevant. The use of biopolymers, materials for creating industrially important plastics and methods for utilizing procedures in this way is fully included in the sustainable development of civilization in general and the polymer industry in particular. From this point of view, the work of Aleksandra Dzejkala, Miroslava Proczon and Natalia Sendzikowska is still relevant.
However, the relevance of the introduction raises a number of questions. This topic is very rapidly developed and the presence of only 8 references out of 33 to a current publication (less than 5 years old) is not enough for a competent literature review. The introduction part needs to be strengthened.
In addition, DSC research is included in the synopsis of the work and the experimental part. But after reading the work several times, I did not see a single DSC curve (the ESI section is missing). The data is shown only in the table, and, to be honest, it is confusing; the errors in temperature measurement in the Tg Onset and Tg Midpoint ISO columns are especially similar. Having carried out many experiments to determine the glass transition temperature, I have never had a situation where, in the presence of components, measurement errors were made.
The work is overloaded with drawings, for example, it is better to transfer faster NMR to the ESI section, where the drawings repeat the data given in the table. By the way, NMR is well discussed, but the ratios of the integral nitintensities of key fragments of macromolecules are not discussed.
There is also no data on the molecular weight distribution of the new polymers obtained, and this is one of the most important characteristics.
Figure 17 discusses panels A and B, but actually shows 4 panels that are not labeled. I understand that this is a DTG curve, but people not familiar with the method may not always understand it. In addition, TG/DTG are created on one YYX type chart, for greater information content.
In section 2.9.7. Microbial aging is based on experimental data. Only a description of experiments and standard samples.
Why is there no assessment of dynamic-mechanical characteristics? The DMA method allows the degradation of polymers to be assessed much more accurately.
In the methods section, the type of registration and spectra is not specified: lumen, diffuse reflection or ATR. Given that the authors couple signals between amide groups, it is necessary to control the thickness of the enhancing layer. However, the spectra do not show the immersion medium, and the sample preparation does not contain a description of obtaining samples of such thickness.
Discussion of microscopy is uninformative, since there is no description of sample preparation. if samples are obtained at different humidity levels, their morphology will differ. For measuring surface properties, AFM in contact mode is best suited to measure hardness in mH.
The work requires significant improvements.
Author Response
- Thank you for your comments. We have made corrections in accordance with the guidelines
- We fully agree, we have made corrections in the form of supplemented literature reports, which have been edited and included in the review.
- Thank you for pointing out the lack of detailed data from DSC studies.
No DSC curves:
We fully agree that presenting the DSC curves in the paper would be beneficial to better understand the results.
Due to the poor resolution of the graphs generated by the DSC device used and the inability to change the language (from PL to EN), the temperature readings have been included in a table without graphs (graphs below in the version from the Mettler Toledo camera).
Unfortunately, I can not attach the file containing the graphs of DSC curves in response. Once again, I emphasize that the thermal analysis apparatus has recorded the curves in Polish, their formatting degrades the quality, I can send these curves to the e-mail, if there is such a need, I will ask for an e-mail.
Confusing data in the table:
I understand that the measurement errors in the Tg Onset and Tg Midpoint ISO columns shown in the table seem incredibly close. However, it should be clarified that these values come from the measurement error of the DSC instrument itself and not from statistical error. The technical documentation of the device specifies the temperature measurement error for the glass transition temperature to be ±2°C. Unfortunately, due to time and material limitations, only one sample could be analyzed at this stage of the research. In further work, it is planned to repeat the DSC tests on a larger number of samples, which will allow for a more accurate determination of the glass transition temperature and estimation of the static error.
- Due to the complexity of the analytical gelatin-agar or gelatin-carrageenan structures obtained as a result of the synthesis of polymer blends, it was necessary to perform both proton and carbon spectra in order to determine specific signals and their location, which is why such a number of spectra were included (NMR section).
Missing issues regarding the analysis of signals originating from amino acids and their functional groups have been added in the section.
- I fully agree that providing information about the molecular weight distribution is extremely important for the full characterization of blends. Unfortunately, we do not have this data at this stage of the research. This is due to the fact that the molecular weight measuring devices available in our laboratory are not suitable for testing polymers with such a high molecular weight, such as gelatin blends.
The molecular weight distribution of gelatin blends with the addition of agarose or carrageenan is expected to be bimodal. This means that the blends will contain both high molecular weight gelatin molecules and lower molecular weight agarose or carrageenan molecules.
In further work, we plan to precisely determine the molecular weight distribution of gelatin blends. For this purpose, we plan to use the following methods: Gel permeation chromatography (GPC): This method allows the separation of polymer molecules according to their size. GPC will provide information about the average molecular weight of the blend and the molecular weight distribution.
- We agree that TG/DTG charts presented on a single YYX-type chart are more beneficial from the point of view of information content. However, in our case, the charts were generated by specialized software provided by the device manufacturer, which presents the data in a specific format.
We understand your concerns about the comprehensibility of the chart. However, we would like to point out that in our work the graphs are an accurate representation of the data obtained from the Mettler Toledo TGA/DSC device. The original data were presented on four panels, but due to image quality constraints, we decided to include in the publication the images we made in Origin.
- Corrections were made in sections 2.9.7 and 2.9.8 according to the comments, the sections were merged and reacted.
- Thank you for pointing out the lack of assessment of dynamic-mechanical properties. I agree that testing the dynamic-mechanical properties (DMA) using the DMA method is a valuable method for assessing polymer degradation. Unfortunately, at this stage of the research we did not have the opportunity to perform this study.
- The methodology introduced additional information regarding FTIR analysis, the research methods used and the method of preparing samples for testing.
- Thank you for drawing attention to the issue of microscopy description and sample preparation.
The reviewer rightly notes that humidity can affect the morphology of samples. However, this applies to samples that have not undergone a thermostabilization process. In the case of samples subjected to this process, the influence of moisture contained in the air on the morphology is negligible. This is because thermal stabilization removes a significant part of the water from the samples, and the remaining amount is too small to cause significant morphological changes.

Reviewer 3 Report
Comments and Suggestions for Authors
The paper entitled „Gelatine Blends Modified with Polysaccharides: A Potential Alternative to Non-Degradable Plastics” focuses on the modification of gelatine blends with two polysaccharides - agar and carrageenan, to obtain a stable biopolymer coating. Various techniques were used to characterize the resulting bioplastics, including FTIR (Fourier transform infrared spectroscopy), TGA (thermogravimetric analysis)/DSC (differential scanning calorimetry), contact angle measurements and surface energy characterization. The influence of thermal and microbiological degradation on the properties of the blends was also investigated. The topic of the paper is interesting and relevant. The results are nicely illustrated and well explained. However, it seems to me that the results should be compared more thoroughly with data known from the literature. I would like to recommend a major revision of the paper for publication because of the following main issues:
1. The introduction should be elaborated with a focus on the issues addressed in the research.
2. The aim of the paper should be better formulated at the end of section 1.
3. What is the substrate material used in this study?
4. The interpretation of the results from the FTIR spectra of samples should be based on literature data and the appropriate reference should be cited in the text.
5. In Fig. 16, the measured amount of adsorbed water seems statistically unreliable. The same applies to Fig. 21, 22 and 23. How many samples have been tested? In the Methodology section, the description of the determination of water absorption is missing.
6. The numbers and text on the axis of Fig. 17 are too small to be nicely seen.
7. The whole text should be elaborated to have comparisons of the results with other similar studies. The interpretation of the results alone carries less information for the reader.
8. Some sentences should be referred to in the text. For example “In the literature, the glass transition temperature of gelatine varies depending on its origin or different extraction methods.”
9. Although section 2.9.3. is entitled “Photos with an optical microscope”, The caption of Figure 20 is “Scanning electron microscope images of gelatine blends”. The images are obtained by optical microscopy. Moreover, no SEM images are present in the whole text, however, in the Methodology section electron microscopy characterization occurs.
10. The conclusion section can be improved. It sounds more like a summary of the results rather than a conclusion.
11. The reference style should be improved and recent references should be included in the study.
Comments on the Quality of English LanguageNone
Author Response
Thank you for your comments. We have made corrections in accordance with the guidelines.
- We fully agree with the comments and have therefore modified the introduction by supplementing it with a relevant bibliography.
- The target was added at the end of section number 1.
- The preparation was corrected and annotated in Section 2.1 Preparation of the Blends. The substrate is a gelatin-glycerol polymer matrix, which during synthesis was enriched with polysaccharides depending on the proposed system, homogenized, and then thermo-stabilized to remove residual solvent. Subsequently, processing methods leading to the appropriate shape of the produced polymer blends were designed.
- Validation of the results obtained by infrared analysis (FTIR) based on literature data was carried out in accordance with the rightly suggested comment made.
- Measurements were made three times for three samples of a given polymer blend, so the results were averaged. Error bars were added. A correction of the graphs was carried out, and a description on the normalized determination of water absorption was introduced into the methodology
In cross-linked during chemical synthesis, polymer blends, in the initial state of the absorption tests, the value of absorbed water was assumed to be 100% (Figs. 16,23). Hence, further absorption of water in the test materials results in an increase in the weight of the samples.
- Figure number 17 has been corrected.
- Very good point, all of the posted studies have been validated with other complementary literature studies.
- The glass transition temperature is an important parameter characterizing the plastic properties of gelatin. The phenomenon of glass transition means that the viscosity of gelatin increases rapidly, and the material becomes more rigid. This is a second-order phase transformation, and its observation is a sudden change in heat capacity. With this in mind, we have included a passage in the text regarding, "In the case of gelatin, the Tg value can vary depending on the histological origin of the animal tissue," along with a citation of the literature on the subject.
- The entire section 2.9.3 has been revised and corrected. The images were taken with an optical microscope and not SEM.
- The proposals have been revised and corrected.
- New literature reports were introduced to enrich the literature review, citations were introduced throughout the research description, and the style of references in the bibliography was standardized.

Round 2
Reviewer 2 Report
Comments and Suggestions for Authors
The authors took into account and corrected these comments.
As a scientist, I am somewhat confused that the device does not allow exporting measurement results in the form of data sets. But the DSC curves presented in the additional materials actually have an extremely low scatter.
The problem of molecular weight distribution of ultra-high molecular weight polymers is quite common. The use of GPC for such systems is not rational, due to the sealing of the columns after 1 measurement.
From personal experience (I worked with polyacrylamides Mw from 0.5 to 15 million daltons), I recommend that you use the classic method of leaking a solution through a capillary, that is, viscosity.
In general, after the submitted revisions, the article can be accepted for publication.
Author Response
Thank you for your response and understanding on some issues..
We will comply with the comments and, as much as possible, in the future we will use the suggested research technique, which will allow a more favorable presentation of such results .
Reviewer 3 Report
Comments and Suggestions for Authors
The authors have addressed most of the reviewer's comments. However, there are still some comments with missing references (such as lines 214-227), and the caption of Figure 20 still contains a mistaken explanation.
Author Response
Thank you for your reply and we apologize for the shortcomings.
Yes indeed an oversight with the caption of figure number 20, we have already corrected the error, in the research methodology the wrong notation of the microscope in question was previously changed.
The literature in the paragraph on infrared spectroscopic studies contained in lines 214-217 has also been corrected.
